# Adsorption of Lead (II) from Aqueous Solution with High Efficiency by Hydrothermal Biochar Derived from Honey

**DOI:** 10.3390/ijerph17103441

**Published:** 2020-05-15

**Authors:** Bo Wang, Jie Yu, Hui Liao, Wenkun Zhu, Pingping Ding, Jian Zhou

**Affiliations:** 1State Key Laboratory of Environment-Friendly Energy Materials, School of Life Science and Engineering, Southwest University of Science and Technology, Mianyang 621010, China; wangbo@live.cn (B.W.); yujie23235@yeah.net (J.Y.); liaohui951020@163.com (H.L.); Zhuwenkun@swust.edu.cn (W.Z.); 2College of Nuclear Technology and Automation Engineering, Chengdu University of Technology, Chengdu 610059, China; dingpp@cdut.edu.cn

**Keywords:** honey, hydrothermal carbon, adsorption, water pollution, lead (II)

## Abstract

A novel natural honey hydrothermal biochar (HHTB) was prepared using natural honey as raw material. The as-prepared adsorbent was applied to adsorb Pb^2+^ from aqueous solution and characterized by scanning electron microscopy, Fourier transform infrared spectroscopy, and X-ray photoelectron spectroscopy to investigate the structure and morphology change of the adsorbent before and after Pb^2+^ adsorption. The influence of the pH, initial Pb^2+^ concentration, temperature, and contact time on the adsorption of Pb^2+^ was systematically investigated. The results revealed that the adsorption capacity for Pb^2+^ is up to 133.2 mg·g^−1^ at initial pH of 5.0 and adsorption temperature of 298 K. Meanwhile, the adsorption of Pb^2+^ on HHTB can be well fitted by the pseudo-second-order model and Langmuir isotherm model. The adsorbent had great selectivity for Pb^2+^ from the aqueous solution containing coexisting ions including Cd^2+^, Co^2+^, Cr^3+^, Cu^2+^, Ni^2+^ and Zn^2+^. Furthermore, the adsorption of Pb^2+^ on HHTB was attributed to complexation coordination, where it involved hydroxyl and carboxylic groups on HHTB in the process of adsorption of Pb^2+^.

## 1. Introduction

In recent years, with the rapid development of industrialization and urbanization, a considerable part of heavy metals are directly discharged into nature and cause severe environmental pollution all over the world. Heavy metals can be enriched in biological organs through the food chain, thus endangering human health [1,2] and breaking the balance of the ecological environment [3]. Lead (Pb) is a kind of heavy metal that may cause serious harm to the human body through affecting the function of the human nervous system, especially Pb to physical development disorders of children [4]. According to the survey, since 1994 the concentration of Pb in Chinese children’s blood has dropped significantly from 92.90 μg L^−1^ to 63.15 μg L^−1^ but is still much higher than that of U.S. children in 2010 (13 μg L^−1^). Pb is widely found in industrial wastewater and sludge such as pigments, textiles and electroplating. Especially in the wastewater discharged from the lead-acid battery industry, the Pb content reaches 200–500 mg L^−1^, which is much higher than that of Pb and zinc industrial pollutants (1 mg L^−1^) and drinking water standards (0.01 mg L^−1^). Therefore, the treatment of Pb pollution in wastewater has attracted much attention for scientists. In recent years, domestic and foreign scholars have conducted a lot of research and developed a variety of effective treatment techniques, such as chemical precipitation [5], ion exchange [6], electrolysis [7], biosorption [8,9,10,11], membrane separation [12], photo reduction and other treatment methods [13,14,15,16]. Biosorption has attracted more and more attention because of its wide source of raw materials, low cost and environmental friendliness [17,18]. Biochar is a highly aromatic solid material obtained by pyrolysis and carbonization of biomass materials under a certain temperature [19]. It has excellent characteristics of good pore structure, rich oxygen-containing functional groups on the surface [20] and strong ion exchange [21] to possess good physical and chemical adsorption capacity [22,23,24,25,26]. Biochar is produced through four main thermochemical routes: (i) torrefaction, (ii) pyrolysis, (iii) hydrothermal carbonization and (iv) gasification and has wide applications, such as soil amendment, environmental remediation, energy storage, composites, and catalyst production [27,28]. In these thermochemical methods, hydrothermal carbonization is useful for the improvement of properties of products [29]. The raw material for making biochar has always been a hotspot for researchers. Honey, as a natural renewable resource, has a structure rich in hydroxyl groups, and the hydroxyl groups of biochar play an important role in its application. According to a survey, the world’s honey output was about 1.863 million tons in 2019, with a steadily increasing trend. The more mature the honey production process, the lower the price will follow. Furthermore, there is much more waste in honey factories, which is relatively cheap. The price of honey is indeed more expensive than some other biological precursor materials, but it is also one of the potential precursors for making biochar.

In this work, a novel natural honey hydrothermal biochar (HHTB) was prepared using natural honey as raw material. The as-prepared adsorbents were characterized by scanning electron microscopy (SEM), Fourier transform infrared (FTIR) spectrum, energy dispersive X-ray (EDX) detection, and X-ray photoelectron spectroscopy (XPS) to investigate the structure and morphology change of the adsorbent before and after Pb^2+^ adsorption. In addition, the effect of various experimental parameters of Pb^2+^ adsorption from aqueous including pH of the solution, co-existing ions, contact time, initial concentration, and temperature, as well as adsorption kinetics, isotherm models, and thermodynamics were discussed.

## 2. Materials and Methods

### 2.1. Experiment Materials

All reagents used in this study were of analytical grade. Honey was purchased from Shanghai Guanshengyuan Bee Products Co, Ltd. (Shanghai, China). All reagents including HCl, HNO_3_, NaOH, the standard Pb^2+^ solution and other metal nitrates used in this work were of analytical grade and obtained from Chengdu KeLong Chemical Co., Ltd., Chengdu, China.

### 2.2. Preparation of HHTB

The 50 g anhydrous honey was added into a beaker containing 100 mL of deionized water and placed on a magnetic stirrer for even stirring. Then, the honey solution was transfer to a Teflon-lined stainless-steel autoclave with a volume capacity of 200 mL. After that, the autoclave was sealed and tempered at 453 K for 24 h and cooled to room temperature. The products were filtered off and washed with deionized water several times, and finally dried at 323 K in a vacuum oven. The obtained hydrothermal honey biochar was placed in a muffle furnace for further carbonization at 573 K for 20 h [22], and the HHTB was obtained.

### 2.3. Characterization of HHTB

The surface morphology and element composition of the HHTB were observed by scanning electron microscopy (SEM) complimented with energy-dispersive X-ray detection (EDX) (Ultra 55, Carl Zeiss, Oberkochen, Germany). A Thermogravimetric Analyzer (TGA, Q500, New Castle, DE, USA) was employed for analysis of the thermal properties of the as-prepared HHTB. The structural information of the samples was recorded by Laser Raman Spectrometer (LRS, Renishaw, London, UK). Fourier transform infrared spectroscopy (FTIR) (Nicolet-5700, PerkinElmer Instruments Corp., Waltham, MA, USA) was employed to measure the change of functional groups of HHTB before and after Pb^2+^ adsorption in the wave number range of 400–4000 cm^−1^. The binding energies before and after Pb^2+^ adsorption were investigated by X-ray photoelectron spectroscopy (XPS) (Thermo Scientific Escalab 250, Thermo Fisher Corp., Waltham, MA, USA). Peaks from all of the high-resolution core spectra were fitted with XPSPEAK 4.1 software.

### 2.4. Batch Adsorption

In this paper, the adsorption of Pb^2+^ on HHTB was investigated for different factors such as pH, ionic strength, contact time, initial concentration and temperature. The adsorption experiment was performed in a reciprocating water bath shaker at a shaking speed of 200 rpm. A total of 0.01 g of HHTB was added to a conical flask that contained 50 mL of a Pb^2+^ solution at a given pH. The pH of the solution was adjusted by 0.05 mol·L^−1^ HNO_3_ or 0.05 mol·L^−1^ NaOH solutions. Then, the mixtures were agitated in a shaking apparatus with concussion speed of 150 rpm at the investigated temperature for a predetermined time. After that, solid−liquid separation was achieved by filtration, and the residual Pb^2+^ concentrations of the filtrates were analyzed by atomic absorption spectroscopy (AAS) method (Z-2300, HITACHI, Japan). The adsorption capacity (Qe) and the removal rate (RE)% are calculated by the following equations:Q_e_ = (C_0_ − C_e_)V/W(1)
RE(%) = (C_0_ − C_e_)/C_0_ × 100(2)
where C_0_ (mg·L^−1^) and C_e_ (mg·L^−1^) are the concentration of Pb^2+^ before and after adsorption, respectively. V (mL) is the volume of the solution and W (g) is the weight of the HHTB. Qe (mg·g^−1^) is the adsorption capacity of HHTB for Pb^2+^ during equilibrium. RE (%) is the removal rate of Pb^2+^ by HHTB during equilibrium.

All experiments mentioned above were carried out in triplicate tests, and the reported results were the average value of three data sets.

## 3. Results and Discussion

### 3.1. Characterization of HHTB

HHTB, with a diameter from 2 μm to 15 μm (Figure 1A), was spherical and easy to reunite. After soaking in the Pb ion solution, it was found that Pb was successfully adsorbed on HHTB (Figure 1B). FT-IR spectra were employed to investigate the changes of functional groups and the results are shown in Figure 2A. It can be seen that the broad adsorption peak at 3430 cm^−1^ (a) was owing to the stretching vibration of hydroxyl, and the peak at about 2800 cm^−1^ may be due to the stretching vibration of -CH_2_-. Meanwhile, the characteristic IR adsorption peak at 1630 cm^−1^ and 1100 cm^−1^ could be a result of the carbonyl stretching vibrations and carboxylic acid bending vibration, respectively [30,31]. What is more, from Figure 2B it can see that there was only one obvious peak, and both of them were around 1600 cm^−1^. Compared with Figure 2A, it is further demonstrated that it may be a carbonyl group. These results indicated that the as-prepared samples were mainly composed of carbon, hydrogen and oxygen and were rich in carbon-oxygen functional groups, which may be the binding sites of heavy metal ions [32] (Figure 2A,B). After adsorption Pb^2+^, the stretching vibration peak of C=O moved to 1620 cm^−1^, indicating the site of the carboxyl functional group (-COOH) on the HHTB was the main adsorption reaction site.

The TGA curves of samples with different treatments—calcined and hydrothermal—are presented in Figure 2C. The water loss rate of the calcined samples was significantly higher than that of the hydrothermal samples while the temperature raised to 100 °C, indicating that the hydrophilic functional groups such as carboxyl groups of the calcined samples increased. Meanwhile, the weight change for the calcined sample was only 5% in the temperature range of 85 °C to 401 °C, but involved much higher weight loss (ca, 23%) for the hydrothermal sample. These phenomena showed that the calcined sample has a clear stabilization stage when the temperature is below 400 °C. Thus, the calcined sample is much more stable than the hydrothermal one, probably due to a more stable substance being formed during the process of solid phase reaction at a high temperature.

In order to understand the binding site of Pb^2+^ on the surface of HHTB, the XPS spectra for wide scan, C 1s and O 1s were recorded. From the wide scan data of HHTB-Pb (Figure 2D), the appearance of the peak of Pb 4f at 143.78 eV and 138.88 eV was roughly the same as Pb 4f7/2 and Pb 4f5/2 in the literature [33] and further confirms that Pb^2+^ was adsorbed on the surface of HHTB successfully. The C 4f peak at binding energy 288.99 eV (Figure 2E) and 286.16 eV attributed to the C=O and C-O of HHTB, respectively [34,35], and they shifted to 288.30 eV and 286.16 eV after adsorption of Pb^2+^. Therefore, it can be inferred that the carboxyl and hydroxyl may be the main binding site for Pb^2+^ in the process of adsorption. Furthermore, the O 1s peak at binding energy 535.55 eV and 532.27 eV was corresponding to the C-O and C=O, respectively [36], and the binding energy shifted to 533.70 eV and 531.72 eV, respectively (Figure 2F). Thus, it can be confirmed that the active groups, including C-O and C=O, were involved to form a stable complexation with the positive Pb^2+^ [37].

### 3.2. Bath Adsorption

Effect of pH. The pH value of the solution is one of the most important factors in the process of adsorption. Generally, the effect of pH value to the adsorption capacity of the adsorbent will be investigated firstly by a series of experiments with different pH values. The adsorption capacity of Pb^2+^ of HHTB at different pH values ranging from 1 to 5 are listed in Figure 3A. It can be seen that the adsorption capacity of Pb^2+^ onto HHTB increased dramatically with the increasing pH value, ranging from 1.0 to 5.0. There was a maximum value (49 mg·g^−1^) at the pH of 5.0. At the lower pH value, the surface of adsorbents was easily protonated and the interaction with Pb^2+^ was weakened via electrostatic repulsion between the protonated adsorbent and the positively charged Pb^2+^. Meanwhile, after the pH value exceeds 5, Pb^2+^ is prone to precipitation, which is not conducive to adsorption and removal. Hence, all of the following experiments were carried out at pH of 5.0.

Effect of adsorbent amount. Generally speaking, the adsorbent dose has a great influence on adsorption capacity and removal rate for the target pollutant. Therefore, the effect of HHTB dose on Pb^2+^ adsorption was investigated using different HHTB amounts ranging from 0.1 g L^−1^ to 0.7 g L^−1^. As listed in Figure 3B, the adsorption amount of Pb^2+^ decreased dramatically with the increasing of the HHTB dose but increased for the Pb^2+^ removal rate. The adsorption capacity for Pb^2+^ decreased to 22.4 mg·g^−1^ when the adsorbent increased to 0.7 g·L^−1^; when the amount of HHTB increased to 0.7 g·L^−1^, the removal rate of Pb^2+^ rose to 97% and tended to balance. The increased active positions raised the Pb^2+^ removal rate, but excess reactive positions were unsaturated to interact with Pb^2+^ because of a certain amount of Pb^2+^ in the solution. Thus, the adsorption amount of Pb^2+^ on HHTB decreased with the increasing dose of HHTB. Overall, the adsorbent dose of 0.2 g L^−1^ was used in the following adsorption experiments.

Effect of the initial Pb^2+^ concentration and isotherm model. In order to investigate the effect of the initial Pb^2+^ concentration on the adsorption capacity of HHTB, different Pb^2+^ concentrations ranging from 5 mg·L^−1^ to 240 mg·L^−1^ were prepared to evaluate the adsorption performance. As shown in Figure 3C, the adsorption capacity of Pb^2+^ on HHTB increased sharply with the increasing initial Pb^2+^ concentration from 5 mg·L^−1^ to 60 mg·L^−1^. After that, it increased slowly and the adsorption capacity of Pb^2+^ reached 104.7 mg·g^−1^ while the initial Pb^2+^ concentration increased to 140 mg·L^−1^ at 328.15 K. The raise in adsorption capacity might ascribe to the mass transfer balance between the high Pb^2+^ concentration in solution and adsorbent. Table 1 shows the Pb^2+^ adsorption capacity of HHTB compared with other adsorbents that have been reported. It was worth noting that the adsorption capacity of HHTB is higher than that of most of adsorbents listed in Table 1, indicating an efficient and potential application for Pb^2+^ removal.

In order to further understand the adsorption mechanism, different isotherm models including Langmuir and Freundlich have been fitted based on the adsorption data (Figure 4) using following equations.
(3)$$\frac{C_{e}}{Q_{e}}=\frac{1}{Q_{m}\times K_{L}}+\frac{C_{e}}{Q_{m}}$$
(4)$$Q_{e}=\frac{K_{L}Q_{m}C_{e}}{1+K_{L}C_{e}}$$
(5)$$\ln Q_{e}=\ln K_{f}+\frac{1}{n}\ln C_{e}$$
(6)$$Q_{e}=K_{f}{C_{e}}^{\frac{1}{n}}$$
where Q_m_ (mg·g^−1^) and Q_e_ (mg·g^−1^) are the maximum adsorption amount of HHTB to Pb^2+^ and the adsorption capacity of HHTB for Pb ions at equilibrium, respectively. Ce (mg·g^−1^) is the concentration of Pb^2+^ in solution after adsorption equilibrium. K_L_ (L·mg^−1^) is the Langmuir equation constant. K_f_ and N are empirical constants of the Freundlich equation.

The fitting parameters are listed in Table 2 and the Linear and nonlinear fit of the Langmuir and Freundlich model are presented in Figure 4. The Langmuir and Freundlich model parameters of Pb^2+^ adsorption by HHTB showed that the maximum value of R^2^ for Freundlich adsorption model was 0.9608 at the temperature range from 288 K to 328 K, while the maximum value of R^2^ for Langmuir model was 0.9897. It means that the Pb^2+^ adsorption process on HHTB was more satisfied to monolayer adsorption on the surface of HHTB. After calculation, the maximum adsorption capacity of this experiment was 107.17 mg·g^−1^. Comparing with Table 1, it can be seen that the maximum adsorption capacity of Pb^2+^ in HHTB is significantly higher than that of other similar biomass materials.

Effect of the contact time on the adsorption and adsorption kinetics. In order to evaluate the adsorption speed of HHTB for Pb^2+^, the effect of the contact time on HHTB was considered and the results are listed in Figure 3D. It was observed that the adsorption capacity of HHTB for Pb^2+^ increased gradually until a plateau occurred within 800 min. This increasing trend indicated that the adsorption capacities of Pb^2+^ at different contact times were corresponding to the number of active sites on HHTB. Furthermore, the adsorption kinetics data of Pb^2+^ ions were fitted by Lagrangian kinetics model [43,44], pseudo-first order kinetic model (7), nonlinear equation (8), pseudo-second order kinetic model (9) and nonlinear Equation (10) (Figure 5). The results were calculated and shown in Table 3.
(7)$$\ln\left(Q_{e}-Q_{t}\right)=\ln Q_{e}-K_{1}t$$
(8)$$Q_{t}=Q_{e}\left(1-e^{-K_{1}t}\right)$$
(9)$$\frac{t}{Q_{t}}=\frac{1}{K_{2}Q_{e}^{2}}+\left(\frac{1}{Q_{e}}\right)t$$
(10)$$Q_{t}=\frac{K_{2}{\mathrm{tQ}}_{e}^{2}}{1+K_{2}{\mathrm{tQ}}_{e}^{2}}$$
where K_1_ (min^−1^) represents the rate constant of the pseudo-first order kinetic equation and K_2_ (g· mg^−1^· min^−1^) is the rate constant of the pseudo-second order kinetic equation. Q_e_ (mg·g^−1^) is the adsorption capacity of HHTB for Pb^2+^ during equilibrium. Q_t_ (mg·g^−1^) is the adsorption capacity of HHTB to Pb^2+^ at a certain time.

From the fitted data listed in Table 2, it can be seen that both the R^2^ of the pseudo-first order and pseudo-second order linear kinetic model were higher than 0.99 in the initial Pb^2+^ concentration of 20 mg·L^−1^ at 298 K. However, the R^2^ value of the pseudo-second order nonlinearity was closer to 0.99. In addition, the calculated adsorption capacity of the pseudo-second-order kinetic model was closer to the experimental value. Thus, it can be recognized that the chemical adsorption process, including the chelated bond between Pb^2+^ and HHTB, was the rate-controlling step.

Adsorption selectivity of HHTB. As we know, Pb^2+^-containing wastewater exists in a complicated system including pigments, textiles and electroplating, in which organics and metal ions coexist. Therefore, in order to develop a new absorbent with practical application value, the selectivity investigation for adsorption is important and necessary. The adsorption selectivity of Pb^2+^ on HHTB from an aqueous solution containing six coexisting metal ions was investigated and the results are listed in Figure 3E. It can be seen that the adsorption ratio of Pb^2+^ on HHTB was much higher than those of other metal ions (Cd^2+^, Co^2+^, Cr^3+^, Cu^2+^, Ni^2+^ and Zn^2+^) [45] in the pH range from 1 to 5. Meanwhile, the adsorption capacity of Pb^2+^ on HHTB increased linearly with the increasing of pH but remained almost unchanged for other coexisting ions. These results indicate that the as-prepared adsorbent had great selectivity for Pb^2+^ from the aqueous solution. As mentioned in FT-IR, for many oxygen functional groups including hydroxyl and carbonyl, the active positions for Pb^2+^ adsorption were generated in the process of carbonization.

Desorption studies: It was found that nitric acid is effective for the desorption of Pb^2+^ from HHTB. The results of desorption experiments showed that the Pb^2+^ on HHTB can be completely desorbed by 0.1 M nitric acid solution. Meanwhile, it was clear from Figure 3F that its adsorption capacity for Pb^2+^ was hardly reduced after recycling four times for the desorbed HHTB, revealing that the as-prepared adsorbent has a good regeneration effect and has great potential to be used as a wide-range, low-cost, reusable biomass adsorbent for the removal of Pb^2+^ from wastewater.

### 3.3. Thermodynamic Analysis

According to the Van’t Hoff equation, the entropy change, enthalpy change of the adsorption reaction and the Gibbs free energy at different temperatures can be obtained. Van’t Hoff equation:(11)$$\Delta G^{0}=-{\mathrm{RTlnK}}_{e}$$
(12)$${\mathrm{lnK}}_{e}=\frac{\Delta S^{0}}{R}-\frac{\Delta H^{0}}{\mathrm{RT}}$$
where ΔG^0^ (kJ ·mol) is the Gibbs free energy; ΔH^0^ (kJ·mol^−1^) is the degeneration and ΔS^0^ (J· mol^−1^· K^−1^) is the entropy change; R (8.314 × 10^−3^ mol^−1^·K^−1^) is general gas constant; T (K) is absolute temperature and Ke is thermodynamic equilibrium constant.

$K_{e}$ is calculated as follows:(13)$$K_{e}=Q_{m}\cdot K_{L}$$
where Qm (mg·g^−1^) is the maximum adsorption amount of HHTB to Pb^2+^ and K_L_ (L·mg-1) is the Langmuir equation constant.

It can be seen from Table 4 that when the temperature is 308 K to 328 K, ΔH^0^ >0, the adsorption is an endothermic process and the high temperature is favorable for the adsorption. When the temperature is ΔG^0^ <0, the adsorption process is spontaneous; with the increase of temperature, ΔG^0^ gradually decreases, indicating that the adsorption experiment will have better adsorption effect under the condition that the relatively higher temperature. In addition, when the temperature is ΔS^0^ >0, this indicates that the randomness of the liquid and solid of the system increases with the increase of temperature when adsorbing. In summary, the adsorption of Pb^2+^ by HHTB is both a spontaneous reaction and a reaction that requires heat absorption.

### 3.4. Preparation and Adsorption Mechanisms

According to the characterization analysis and the fitting of the adsorption model, a likely adsorption mechanism should be attributed to the hydroxyl and carboxylic groups on HHTB, which results in complexation in the process of adsorption Pb^2+^, as exhibited in Figure 6. Honey contains a lot of monosaccharides, including glucose and fructose. In the hydrothermal process, hydroxymethylfurfural and furanal are easily produced as intermediate products and eventually polymerized to form black spheres. Then, the hydrothermal char is calcined to generate more carboxyl groups and more pores at 573 K. The adsorption of Pb^2+^ on HTTB can be considered as follows: First, two adjacent phenolic hydroxyl or carboxyl groups of HHTB are negatively charged by deprotonation, releasing protons into the solution. Then, a five-membered ring forms through a chelating reaction between one Pb^2+^ and two deprotonated hydroxyl or carboxyl groups. This hypothesized adsorption mechanism can explain the effect of pH on adsorption capacity discussed earlier. At lower pH values, HHTB is easier to protonate which makes the hydroxyl or carboxyl group difficult to chelate with Pb^2+^. As pH value increases, HHTB deprotonation degree increases and reaches a maximum at pH 5.0. Therefore, the hydroxyl and carboxyl groups on HHTB can chelate with Pb^2+^ at pH 5.0 and the adsorption capacity reaches a maximum.

## 4. Conclusions

In this work, a novel natural honey hydrothermal biochar (HHTB) was prepared using natural honey as raw material which was used as an effective adsorbent for removal of Pb^2+^ from an aqueous solution. HHTB has a good adsorption effect on Pb^2+^ with a maximum adsorption capacity of 133.2 mg·g^−1^, and its adsorption process conforms to the pseudo-second-order kinetic model and Langmuir model. Thermodynamics implied that the adsorption process was a spontaneous endothermic reaction. HHTB had great selectivity for Pb^2+^ from the aqueous solution containing coexisting ions of Cd^2+^, Co^2+^, Cr^3+^, Cu^2+^, Ni^2+^ and Zn^2+^. Furthermore, the Pb^2+^ adsorption mechanism on HHTB was mainly attributed to the hydroxyl and carboxylic groups on HHTB, which results in complexation in the process of adsorption of Pb^2+^. Therefore, HHTB has great potential to be a low-cost and high-efficiency adsorbent for Pb^2+^ recovery from an aqueous solution.

## Figures and Tables

**Figure 1 ijerph-17-03441-f001:**
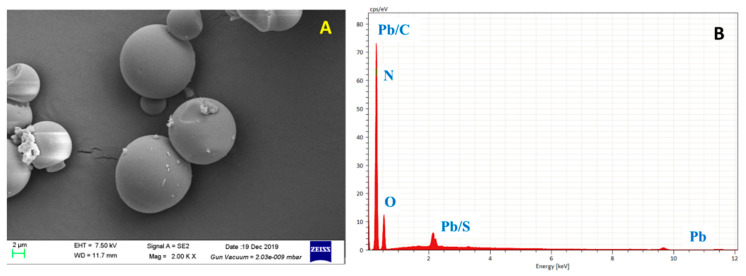
Image (**A**) and EDS (**B**) curve of honey hydrothermal biochar (HHTB) after adsorption of Pb^2+^.

**Figure 2 ijerph-17-03441-f002:**
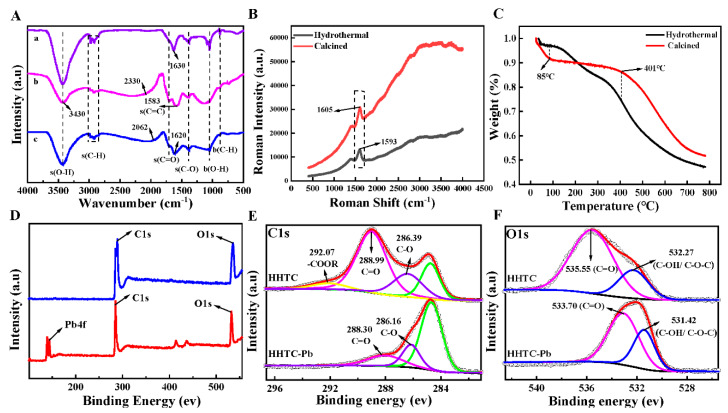
(**A**) FT-IR spectrum before (a) hydrothermal, (b) calcined and after (c) HHTB adsorption of Pb; (**B**) raman analysis of materials before and after HHTB adsorption of Pb; (**C**) thermogravimetric analysis of materials treated with hydrothermal and muffle furnaces; XPS spectrum: (**D**) full spectrum of XPS before and after HHTB adsorption; (**E**) C1s orbital energy level before and after HHTB adsorption; (**F**) O1s orbital energy level before and after HHTB adsorption.

**Figure 3 ijerph-17-03441-f003:**
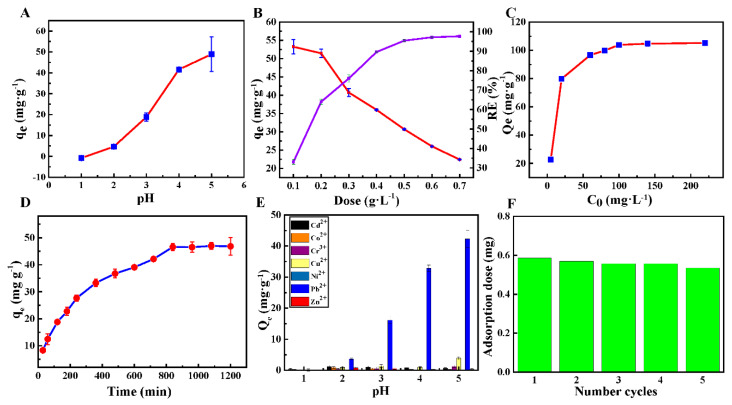
Effect of pH (**A**), adsorbent dose (**B**), concentration of Pb ions (**C**), adsorption kinetics of Pb^2+^ onto HHTB (**D**) and coexisting ions on the adsorption (**E**); HHTB to Pb^2+^ adsorption dose and recycling (**F**).

**Figure 4 ijerph-17-03441-f004:**
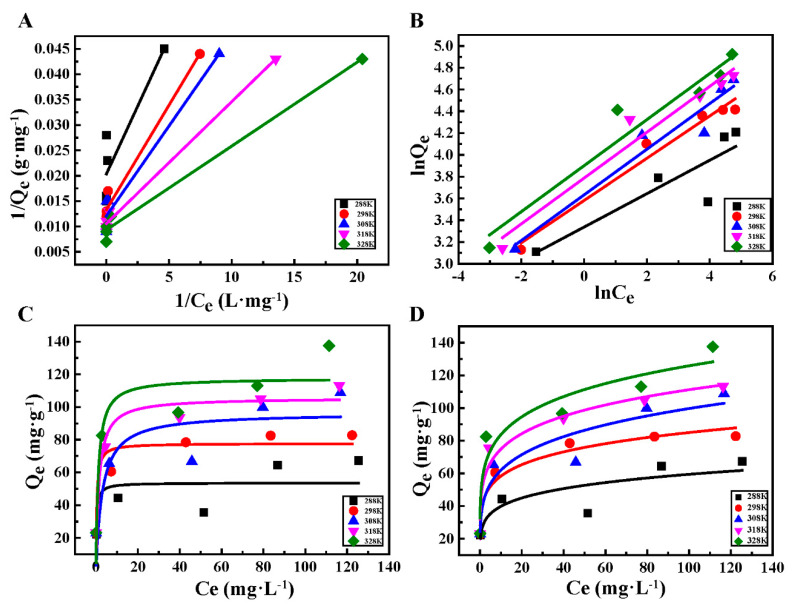
(**A**,**C**) Linear and nonlinear fit of the Langmuir model; (**B**,**D**) linear and nonlinear fits of the Freundlich model.

**Figure 5 ijerph-17-03441-f005:**
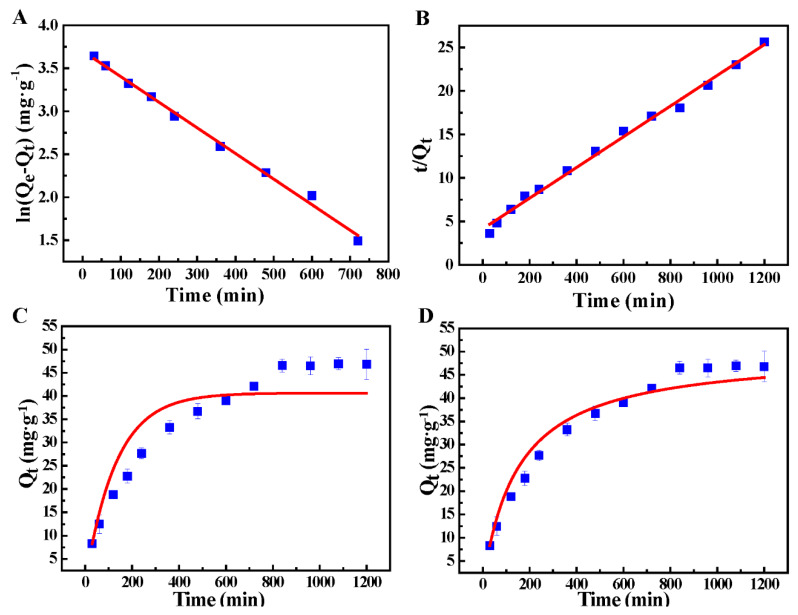
(**A**,**C**) Linear and nonlinear fitting plots of pseudo-first order kinetic models; (**B**,**D**) linear and nonlinear fitting plots of pseudo-second order kinetic models.

**Figure 6 ijerph-17-03441-f006:**
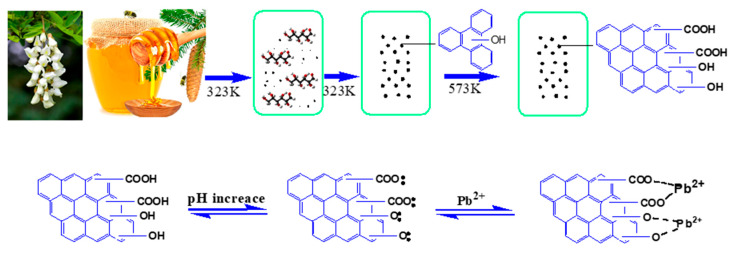
The adsorption mechanism of HHTB.

**Table 1 ijerph-17-03441-t001:** Comparison of Pb^2+^ maximum adsorption capacity (Q _max_) between HHTB and other biomass materials.

Biomass Material	pH	T (K)	Q _max_ (mg·g^−1^)	Literature
Palm Kernel Shell	4.0	298	104.0	[38]
HHTB	5.0	328	107.17	This article
Coconut fiber	—	573	105.5	[39]
Pine cone shell	5.0	298	17.41	[40]
Biopolymer GT	5.0	298	101.74	[41]
Biochar (Coaltec Energy, USA Inc)	5.0	295	37.80	[42]

**Table 2 ijerph-17-03441-t002:** Langmuir and Freundlich model parameters of HHTB adsorption Pb^2+^.

Type	T / K	Langmuir			Freundlich		
		*Q_m_*/(mg·g^−1^)	*K_L_*/(L·mg^−1^)	*R^2^*	*K_f_*	n	*R* ^2^
Linear	288	49.2368	3.7963	0.7525	28.1582	6.5249	0.7015
	298	75.1880	3.2282	0.9762	35.9346	5.1562	0.9608
	308	82.9187	3.3972	0.9557	37.8413	4.7746	0.9233
	318	94.6074	4.4042	0.9897	44.1521	4.7540	0.9453
	328	107.1711	5.6545	0.9848	49.4781	4.7337	0.9210
Nonlinear	288	53.5934	3.0680	0.3709	26.6208	5.7407	0.6375
	298	77.7247	2.5439	0.8770	39.9690	6.1060	0.9447
	308	96.4243	0.3264	0.6488	38.4166	4.8303	0.8643
	318	105.6159	0.7035	0.8745	50.3250	5.8075	0.9467
	328	117.6933	0.8858	0.7989	55.6299	5.6401	0.8985

**Table 3 ijerph-17-03441-t003:** Pseudo-first order kinetics and pseudo-second dynamics simulation parameters of Pb^2+.^in HHTB.

Type	Lagrangian Quasi-First-Order Kinetic Constant			Lagrangian Quasi-Secondary Kinetic Constant		
	K_1_ (min^−1^)	Q_e_ (mg·g^−1^)	R^2^	K_2_ (g·mg^−1^·min^−1^)	Q_e_ (mg·g^−1^)	R^2^
Linear	0.0030	40.4368	0.9946	0.0177	56.6251	0.9933
Nonlinear	0.0076	40.6446	0.9685	1.3272	50.0091	0.9898

**Table 4 ijerph-17-03441-t004:** Thermodynamic parameters of HHTB adsorption of Pb.

T/K	ΔH^0^ (kJ·mol^−1^)	ΔG^0^ (kJ·mol^− 1^)	ΔS^0^ (J·mol^−1^· K^−1^)
288	14.1578	−9.3302	0.0824
298		−10.7031	
308		−11.3129	
318		−12.0288	
328		−12.7474	

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
