# Peer review of "Adsorption of Lead (II) from Aqueous Solution with High Efficiency by Hydrothermal Biochar Derived from Honey"

_ijerph, 2020, doi:10.3390/ijerph17103441_

Round 1

Reviewer 1 Report

The document is well written and the information it presents is interesting.

On line 257

There is an error in the word mechanism

I have questions for the author. If this material were applied to a real water sample, do you know what the pH of the sample is? and how would you adjust the pH to a value of 5 and how feasible is it?

Reviewer 2 Report

It is necessary to present the "state of the art" in the theme, not just present the fundamentals. Honey biochar has already been used in other works, which should be reported in the introduction. It should also be noted that this is not an original contribution (the biocarbon of honey). The original aspect is the application for the removal of lead. However, even the application for lead removal is already mentioned in the literature, as in the paper of Duwiejuah et al. (2017)

The methodology should be much more in-depth. In the preparation of the biochar, the agitation speed in the dissolution of the honey is not indicated. Mass studies need to be more detailed, as well as pH. The point of zero charge (PZC) must be analyzed. The quantification of the adsorption of the metal on the filter paper has not been evaluated

Kinetic, thermodynamic and equilibrium studies, as well as the models used, must be described in the methodology, in a coherent and scientific way, allowing the reproducibility of the results

The characterization for this type of material, necessarily passes through the study of the surface area and the pores by microporosimetry and evaluation of the adsorption isotherms. These characterizations are necessary for the discussion of results and purchase with other adsorbent material

the article is focused only on the study of lead, however in the conclusions it inserts other metals. You have to show the results obtained with the other metals to analyze the affinity of the biochar with the metals

Reviewer 3 Report

1why using honey to prepare biochar, does the honey itself adsorb pb?

2The thermodynamic analysis should be carried out.

3When other ions existed, did the isotherm change? are there any competition? should the competition adsorption isotherm be applied?

Reviewer 4 Report

Heavy metal adsorption by biomaterials is a well-studied story. In recent years, the studies on heavy metal removal by adsorption has become less and less attractive as we have known the mechanisms involved this process. Although the work present here is well-conducted and the manuscript is also well-prepared, the authors have not told us why they did the work and what was the novelty of the story. Moreover, I have also found several major gaps and minor mistakes throughout the paper, which should be carefully revised before consideration for potential publication in IJERPH. The following suggestions are provided for improving their manuscript.

Major gaps:

  1. Line 45. 700 oC is not high in preparation of biochar, but “low temperature” here can confuse readers.
  2. Line 59. Did you use “honey” or “honeycomb”? If you used “honey” to prepare biochar, I think the cost is very high. You need to compare the cost with the previous materials. All organism-derived materials can be used to prepare biochar, I don’t think this idea can make novel findings. You should introduce the material in INTRODUCTION part and let us know why you focus on it. Pls check the similar mistakes throughout the manuscript and correct it one by one, e.g., Line 121.
  3. Line 105. “lead”, pls use lead or Pb2+ throughout the paper, don’t use two.
  4. Lines 107-112. You should cite some reference to support your data, i.e., the functional groups information. Similar gaps are also found in Lines 126-145.
  5. Line 121. Pls show us what abc indicate in Fig. 2A? The legend only gives the information of “FT-IR spectrum before and after HHTB adsorption of lead”. We don’t know which one corresponds to a, b or c.
  6. Line 183. Pls move the table title to the next page to get the title and content together. Similar mistake is also found in Table 3.

Minor mistakes:

  1. Line 87. Pls correct the mistakes of “Pb2+”, “2+” should be given as upper-case. Pls check the similar mistakes throughout the manuscript.
  2. Line 104. “2um to 15um” should be “2 μm to 15 μm”, pls check the similar mistakes throughout the manuscript, e.g., Line 251.
  3. Line 152. “mgg-1” should be “mg g-1”, making it consistent with others. Similar mistakes are also found in Line 162. Pls check the similar mistakes throughout the manuscript.
  4. Line 168. Figs. B & C, pls use a blank space between name and unit. All the similar mistakes should be corrected in the Y axis of all figures. Pls check the similar mistakes throughout the manuscript, e.g., Table 1, Line 194, Fig. 4, etc.
  5. Line 208. The line to separate linear and non-linear was located in wrong space.

Round 2

Reviewer 4 Report

The authors have carefully corrected their manuscript, I think it can be considered for potential publication in IJERPH.